# Unorganized Malicious Attacks Detection

**Ming Pang**    **Wei Gao**    **Min Tao**    **Zhi-Hua Zhou**
National Key Laboratory for Novel Software Technology,
Nanjing University, Nanjing, 210023, China
{pangm, gaow, zhouzh}@lamda.nju.edu.cn   taom@nju.edu.cn

## Abstract

Recommender systems have attracted much attention during the past decade. Many attack detection algorithms have been developed for better recommendations, mostly focusing on shilling attacks, where an attack organizer produces a large number of user profiles by the same strategy to promote or demote an item. This work considers another different attack style: *unorganized malicious attacks*, where attackers individually utilize a small number of user profiles to attack different items without organizer. This attack style occurs in many real applications, yet relevant study remains open. We formulate the unorganized malicious attacks detection as a matrix completion problem, and propose the Unorganized Malicious Attacks detection (*UMA*) algorithm, based on the alternating splitting augmented Lagrangian method. We verify, both theoretically and empirically, the effectiveness of the proposed approach.

## 1    Introduction

Online activities have been an essential part in our daily life as the flourish of Internet, and it is important to recommend suitable products effectively as the number of users and items increases drastically. Various collaborative filtering techniques have been developed in diverse systems to help customers choose their favorite products in a set of items [5, 18, 28]. However, most collaborative filtering approaches are vulnerable to spammers and manipulations of ratings [13, 19], and attackers could bias systems by inserting fake rating scores into the user-item rating matrix. Some attackers try to increase the popularity of their own items (push attack) while the others intend to decrease the popularity of their competitors' items (nuke attack).

Detecting attacks from online rating systems is crucial to recommendations. Most attack detection studies focus on shilling attacks [13], where all the attack profiles are produced by the same strategy to promote or demote a particular item. For example, an attack organizer may produce hundreds of fake user profiles with one strategy where each fake user profile gives high scores to the most popular movies and low scores to the target movie. Relevant studies have shown good detection performance on diverse shilling attack strategies [16, 19, 23].

Practical mechanisms have been developed to prevent shilling attacks. For example, lots of online sites require real names and phone numbers for user registration; CAPTCHA is used to determine whether the response is generated by a robot; customers are allowed to rate a product after purchasing this product on the shopping website. These mechanisms produce high cost for conducting traditional shilling attacks; for example, small online sellers in e-commerce like Amazon have insufficient capacity to produce hundreds of fake rating profiles to conduct a shilling attack.

In this paper, we introduce another different attack model named *unorganized malicious attacks*, where attackers individually use a small number of user profiles to attack their own targets without organizer. This attack happens in many real applications: online sellers on Amazon may produce a few fake customer profiles to demote their competitors' high-quality products; writers may hire

several users to give high scores to promote their own books. Actually, recommender systems may be seriously influenced by small amounts of unorganized malicious attacks, e.g., the first maliciously bad rating can decrease the sales of one seller by 13% [20]. So far as we know, the detection of unorganized malicious attacks has rarely been studied, and existing attack detection approaches do not work well on this kind of attack [26].

We formulate the unorganized malicious attacks detection as a variant of matrix completion problem. Let $X$ denote the ground-truth rating matrix without attacks and noises, and assume that the matrix is low-rank since the users' preferences are affected by several factors [31]. Let $Y$ be the sparse malicious-attack matrix, and $Z$ denotes a small perturbation noise matrix. What we can observe is a matrix $M$ such that $M = X + Y + Z$.

We propose the Unorganized Malicious Attacks detection (*UMA*) algorithm, which can be viewed as an extension of alternating splitting augmented Lagrangian method. Theoretically, we show that the low-rank rating matrix $X$ and the sparse matrix $Y$ can be recovered under some classical matrix-completion assumptions, and we present the global convergence of UMA with a worst-case $O(1/t)$ convergence rate. Finally, empirical studies are provided to verify the effectiveness of our proposed algorithm in comparison with the state-of-the-art methods for attack detection.

The rest of this paper is organized as follows. Section 2 reviews related work. Section 3 introduces the framework of unorganized malicious attacks detection. Section 4 proposes the UMA algorithm. Section 5 shows the theoretical justification. Section 6 reports the experimental results. Section 7 concludes this work.

## 2 Related Work

Collaborative filtering has been one of the most successful techniques to build recommender systems. The core assumption of collaborative filtering is that if users have expressed similar interests in the past, they will share common interest in the future [12]. Significant progress about collaborative filtering has been made [5, 18, 28, 31]. There are two main categories of conventional collaborative filtering (based on the user-item rating matrix) which are memory-based and model-based algorithms.

Collaborative filtering schemes are vulnerable to attacks [1, 13], and increasing attention has been paid to attack detection. Researchers have proposed several methods which mainly focus on shilling attacks where the attack organizer produces a large number of user profiles by the same strategy to promote or demote a particular item. These methods mainly contain statistical, classification, clustering and data reduction-based methods [13].

Statistical methods are used to detect anomalies with suspicious ratings. Hurley *et al.* [16] proposed the Neyman-Pearson statistical attack detection method to distinguish malicious users from normal users, and Li and Luo [17] introduced the probabilistic Bayesian network models. Based on attributes derived from user profiles, classification methods detect attacks by kNN, SVM, etc. [14, 24]. Bhaumik *et al.* [3] presented the unsupervised clustering algorithm based on several classification attributes [7], and they apply $k$-means clustering based on these attributes and classify users in the smallest cluster as malicious users. Variable selection method treats users as variables and calculates their covariance matrix [22]. Users with the smallest coefficient in the first $l$ principal components of the covariance matrix are classified as malicious users. Ling *et al.* [19] utilized a low-rank matrix factorization method to predict the users' ratings. Users' reputation is computed according to the predicted ratings and low-reputed users are classified as malicious users.

These methods make detection by finding the common characteristics of the attack profiles that differ from the normal profiles. Therefore, they have a common assumption that the attack profiles are produced by the same attack strategy. However, this assumption does not hold for unorganized malicious attacks, where different attackers use different strategies to attack their own targets.

Recovering low-dimensional structure from a corrupted matrix is related to robust PCA [4, 9, 33]. However, robust PCA focuses on recovering low-rank part $X$ from complete or incomplete matrix, and the target is different from attacks detection (which is our task). Our work considers the specific properties of malicious attacks to distinguish the attack matrix $Y$ from the small perturbation noise term $Z$. In this way, our method can not only recover the low-rank part $X$, but also distinguish $Y$ from the noise term $Z$ which leads to better performance.

# 3 The Formulation

This section introduces some notations and problem formulation. We introduce the general form of an attack profile, and give a detailed comparison between unorganized malicious attacks and shilling attacks, followed by the corresponding detection problem formulation.

## 3.1 Notations

We begin with some notations used in this paper. Let $\|X\|$, $\|X\|_F$ and $\|X\|_*$ denote the operator norm, Frobenius norm and nuclear norm of matrix $X$, respectively. Let $\|X\|_1$ and $\|X\|_\infty$ be the $\ell_1$ and $\ell_\infty$ norm of matrix $X$, respectively. Further, we define the Euclidean inner product between two matrices as $\langle X, Y \rangle := \text{trace}(XY^\top)$, where $Y^\top$ means the transpose of $Y$. We have $\|X\|_F^2 = \langle X, X \rangle$.

Let $P_\Omega$ denote an operator of linear transformation over matrices space, and we also denote by $P_\Omega$ the linear space of matrices supported on $\Omega$ when it is clear from the context. Then, $P_{\Omega^\top}$ represents the space of matrices supported on $\Omega^c$. For an integer $m$, let $[m] := \{1, 2, \ldots, m\}$.

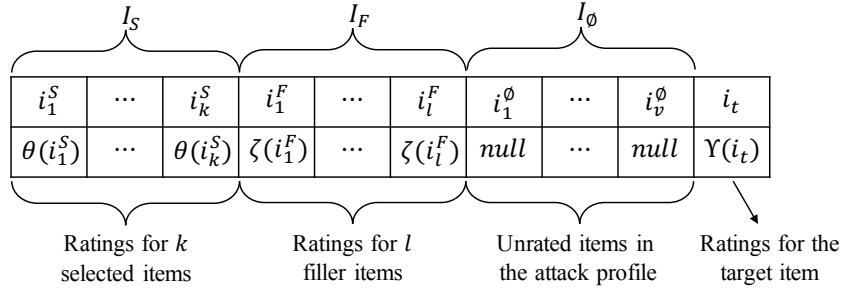

Figure 1: General form of an attack profile.

## 3.2 Problem Formulation

Bhaumik *et al.* [2] introduced the general form of an attack profile, as shown in Figure 1. The attack profile contains four parts. The single target item $i_t$ is given a malicious rating, i.e., a high rating in a push attack or a low rating in a nuke attack. The selected items $I_S$ are a group of selected items for special treatment during the attack. The filler items $I_F$ are selected randomly to complete the attack profile. The null part $I_\emptyset$ contains the rest of the items with no ratings. Functions $\theta$, $\zeta$ and $\Upsilon$ determine how to assign ratings to items in $I_S$, $I_F$ and target item $i_t$, respectively. Three basic attack strategies are listed as follows.

- Random attack: $I_S$ is empty; $I_F$ is selected randomly, and function $\zeta$ assigns ratings to $I_F$ by generating random ratings centered around the overall average rating in the database.
- Average attack: $I_S$ is empty; $I_F$ is selected randomly, and function $\zeta$ assigns ratings to $I_F$ by generating random ratings centered around the average rating of each item.
- Bandwagon attack: $I_S$ is selected from the popular items and function $\theta$ assigns high ratings to $I_S$. The filler items $I_F$ are handled similarly to random attack.

The shilling attack chooses one attack strategy (e.g., average attack strategy), and fixes the target item $i_t$, the numbers of rated items $k$ and $l$ and the rating functions. This makes the generated attack profiles have some common characteristics in one shilling attack. Besides, a large number of attack profiles are required in the basic setting of shilling attacks.

However, unorganized malicious attacks allow the concurrence of various attack strategies, and the number of rated items, the target item and the rating functions can be different. Each attacker produces a small number of attack profiles with their own strategies and preference [26].

Let $U_{[m]} = \{U_1, U_2, \ldots, U_m\}$ and $I_{[n]} = \{I_1, I_2, \ldots, I_n\}$ denote $m$ users and $n$ items, respectively. Let $X \in \mathcal{R}^{m \times n}$ be the ground-truth rating matrix. $X_{ij}$ denotes the score that user $U_i$ gives to item $I_j$ without any attack or noise, i.e., $X_{ij}$ reflects the ground-truth feeling of user $U_i$ on item $I_j$. Suppose that the score range is $[-R, R]$, and we have $-R \leq X_{ij} \leq R$. In this work, we assume that $X$ is a

low-rank matrix as in classical matrix completion [30] and collaborative filtering [31]. The intuition is that the user' preferences may be influenced by a few factors.

The ground-truth matrix $X$ may be corrupted by a system noisy matrix $Z$. For example, if $X_{ij} = 4.8$ for $i \in [m]$, then, it is acceptable that user $U_i$ gives item $I_j$ score 5 or 4.6. In this paper, we consider the independent Gaussian noise, i.e., $Z = (Z_{ij})_{m \times n}$ where each element $Z_{ij}$ is drawn i.i.d. from the Gaussian distribution $\mathcal{N}(0, \sigma)$ with parameter $\sigma$.

Let $M$ be the observed rating matrix. We define the *unorganized malicious attacks* formally as follows: for every $j \in [n]$, we have $|U^j| < \gamma$ with $U^j = \{U_i | i \in [m] \ \& \ |M_{ij} - X_{ij}| \geq \epsilon\}$. The parameter $\epsilon$ distinguishes malicious users from the normal, and parameter $\gamma$ limits the number of user profiles attacking one item. Intuitively, unorganized malicious attacks consider that attackers individually use a small number of user profiles to attack their own targets, and multiple independent shilling attacks can be regarded as an example of unorganized malicious attacks if each shilling attack contains a small number of attack profiles.

It is necessary to distinguish unorganized malicious attacks from noise. Generally speaking, user $U_i$ gives item $I_j$ a normal score if $|M_{ij} - X_{ij}|$ is very small, while user $U_i$ makes an attack to item $I_j$ if $|M_{ij} - X_{ij}| \geq \epsilon$. For example, if the ground-truth score of item $I_j$ is 4.8 for user $U_i$, then user $U_i$ makes a noisy rating by giving $I_j$ score 5, yet makes an attack by giving $I_j$ score $-3$. Therefore, we assume that $\|Z\|_F \leq \delta$, where $\delta$ is a small parameter.

Let $Y = M - X - Z = (Y_{ij})_{m \times n}$ be the malicious-attack matrix. Then, $Y_{ij} = 0$ if user $U_i$ does not attack item $I_j$; otherwise $|Y_{ij}| \geq \epsilon$. We assume that $Y$ is a sparse matrix, whose intuition lies in the small ratio of malicious ratings to all the ratings. Notice that we can not directly recover $X$ and $Y$ from $M$ because such recovery is an NP-Hard problem [9]. We consider the optimization problem as follows:

$$\min_{X,Y,Z} \|X\|_* + \tau \|Y\|_1 - \alpha \langle M, Y \rangle + \frac{\kappa}{2} \|Y\|_F^2$$
$$\text{s.t. } X + Y + Z = M, \ \|Z\|_F \leq \delta. \tag{1}$$

Here $\|X\|_*$ acts as a convex surrogate of the rank function to pursue the low-rank part. $\|Y\|_1$ is used to induce the sparse attack part. The term $\langle M, Y \rangle$ is introduced to better distinguish $Y$ and $Z$, since the malicious rating bias $Y_{ij}$ and the observed rating $M_{ij}$ have the same sign, i.e., $M_{ij} Y_{ij} > 0$, while each entry in $Z$ is small and $Z_{ij} M_{ij}$ can be either positive or negative. We have $Y_{ij} < 0$ and $M_{ij} < 0$ if it is a nuke attack; we also have $Y_{ij} > 0$ and $M_{ij} > 0$ if it is a push attack. So the term $\langle M, Y \rangle$ distinguishes $Y$ from $Z$. $\|Y\|_F^2$ is another strongly convex regularizer for $Y$. This term also guarantees the optimal solution. $\tau$, $\alpha$ and $\kappa$ are tradeoff parameters.

In many real applications, we can not get the full matrix $M$, and partial entries can be observed. Let $\Omega \in [m] \times [n]$ be the set of observed entries. We define an orthogonal projection $P_\Omega$ onto the linear space of matrices supported on $\Omega \subset [m] \times [n]$, i.e.,

$$P_\Omega M = \begin{cases} M_{ij} & \text{for } (i, j) \in \Omega, \\ 0 & \text{otherwise.} \end{cases}$$

The optimization framework for unorganized malicious attack detection can be formulated as follows.

$$\min_{X,Y,Z} \|X\|_* + \tau \|Y\|_1 - \alpha \langle \bar{M}, Y \rangle + \frac{\kappa}{2} \|Y\|_F^2$$
$$\text{s.t. } X + Y + Z = \bar{M}, \ Z \in \mathbf{B}, \ \mathbf{B} := \{Z | \|P_\Omega(Z)\|_F \leq \delta\}, \tag{2}$$

where $\kappa > 0$ and $\bar{M} := P_\Omega(M)$. This formulation degenerates into robust PCA as $\kappa \to 0$ and $\alpha \to 0$. There have been many studies focusing on recovering low-rank part $X$ from complete or incomplete matrix [9, 11, 21, 27], while we distinguish the sparse attack term $Y$ from the small perturbation term $Z$. $\langle \bar{M}, Y \rangle$ is added to find nonzero entries of $Y$, and this yields better detection performance.

## 4  The Proposed Approach

In this section, we propose an alternating splitting augmented Lagrangian method to solve the optimization problem (2), which can be guaranteed with global convergence.

---
**Algorithm 1** The UMA Algorithm
---
**Input:** matrix $M$ and parameters $\tau$, $\alpha$, $\beta$, $\delta$ and $\kappa$.
**Output:** Label vector $[y_1, \dots, y_m]$ where $y_i = 1$ if user $U_i$ is a malicious user; otherwise $y_i = 0$.
**Initialize:** $Y^0 = X^0 = \Lambda^0 = 0$, $y_i = 0$ $(i = 1, \dots, m)$, $k = 0$
**Process:**
 1: **while** not converged **do**
 2:     Compute $Z^{k+1}$, $X^{k+1}$ and $Y^{k+1}$ by Eq. (4), (5) and (6), respectively.
 3:     Update the Lagrange multiplier $\Lambda^{k+1}$ by $\Lambda^k - \beta(X^{k+1} + Y^{k+1} + Z^{k+1} - \bar{M})$.
 4:     $k = k + 1$.
 5: **end while**
 6: if $\max(|Y_{i,:}|) > 0$, then $y_i = 1$ $(i = 1, \dots, m)$.
---

The separable structure emerging in the objective function and constrains in Eq. (2) motivates us to derive an efficient algorithm by splitting the optimization problem. However, it is rather difficult to optimize this problem with theoretical guarantee, because this optimization involves three-block variables. It is well-known that the direct extension of the alternating direction method of multipliers may not be convergent for solving Eq. (2), a three-block convex minimization problem [10, 15, 32].

We propose an alternating splitting augmented Lagrangian method to decompose the optimization of Eq. (2) into three sub-optimizations for the solutions of $Z^{k+1}$, $X^{k+1}$ and $Y^{k+1}$ separately. We will provide global convergence guarantee with a worst-case $O(1/t)$ convergence rate in Section 5.

We first get the augmented Lagrangian function of Eq. (2) as

$$\mathcal{L}_{\mathcal{A}}(X, Y, Z, \Lambda, \beta) := \|X\|_* + \tau\|Y\|_1 - \alpha\langle \bar{M}, Y\rangle + \frac{\kappa}{2}\|Y\|_F^2 - \langle \Lambda, L\rangle + \frac{\beta}{2}\|L\|_F^2, \quad (3)$$

where $L = X + Y + Z - \bar{M}$ and $\beta$ is a positive constant.

Given $(X^k, Y^k, \Lambda^k)$, we update $Z^{k+1}$ with the closed-form solution

$$Z_{ij}^{k+1} = \begin{cases} \min\{1, \delta/\|P_\Omega N\|_F\}N_{ij} & \text{if } (i,j) \in \Omega, \\ N_{ij} & \text{otherwise,} \end{cases} \quad (4)$$

where $N = \frac{1}{\beta}\Lambda^k + \bar{M} - X^k - Y^k$. Lemma 2 gives the closed solution of $X^{k+1}$ as

$$X^{k+1} = \mathcal{D}_{1/\beta}(\bar{M} + \frac{1}{\beta}\Lambda^k - Y^k - Z^{k+1}), \quad (5)$$

where the nuclear-norm-involved shrinkage operator $\mathcal{D}_{1/\beta}$ is defined in Lemma 2. Further, we update $Y^{k+1}$ and Lemma 1 gives the closed solution $Y^{k+1}$ as

$$Y^{k+1} = \mathcal{S}_{\tau\upsilon}(\frac{\alpha+\beta}{\beta}\bar{M} + \frac{1}{\beta}\Lambda^k - Z^{k+1} - X^{k+1})\upsilon\beta, \quad (6)$$

where $\upsilon = 1/(\beta + \kappa)$ and the shrinkage operator $\mathcal{S}_{\tau\upsilon}$ is defined in Lemma 1. Finally, we update

$$\Lambda^{k+1} = \Lambda^k - \beta(X^{k+1} + Y^{k+1} + Z^{k+1} - \bar{M}).$$

The pseudocode of the UMA algorithm is given in Algorithm 1.

## 5 Theoretical Analysis

This section presents our main theoretical results, whose detailed proofs and analysis are given in the supplement document due to the page limitation. We begin with two helpful lemmas for the deviation of our proposed algorithm as follows.

**Lemma 1** *[6] For $\tau > 0$ and $T \in \mathcal{R}^{m \times n}$, the closed solution of $\min_Y \tau\|Y\|_1 + \|Y - T\|_F^2/2$ is matrix $\mathcal{S}_\tau(T)$ with $(\mathcal{S}_\tau(T))_{ij} = \max\{|T_{ij}| - \tau, 0\} \cdot \text{sgn}(T_{ij})$, where $\text{sgn}(\cdot)$ means the sign function.*

**Lemma 2** *[8] For $\mu > 0$ and $Y \in \mathcal{R}^{m \times n}$ with rank $r$, the closed solution of $\min_X \mu\|X\|_* + \|X - Y\|_F^2/2$ is given by $\mathcal{D}_\mu(Y) = S \text{diag}(\mathcal{S}_\mu(\Sigma))D^\top$, where $Y = S\Sigma D^\top$ denotes the singular value decomposition of $Y$, and $\mathcal{S}_\mu(\Sigma)$ is defined in Lemma 1.*

We now present theoretical guarantee that UMA can recover the low-rank component $X_0$ and the sparse component $Y_0$. For simplicity, our theoretical analysis focuses on square matrix, and it is easy to generalize our results to the general rectangular matrices.

Let $X_0 = S\Sigma D^\top = \sum_{i=1}^r \sigma_i s_i d_i^\top$ be the singular value decomposition of $X_0 \in \mathcal{R}^{n \times n}$, where $r$ is the rank of matrix $X_0$, and $\sigma_1, \ldots, \sigma_r$ are the positive singular values, and $S = [s_1, \ldots, s_r]$ and $D = [d_1, \ldots, d_r]$ are the left- and right-singular matrices, respectively. For $\mu > 0$, we assume

$$\max_i \|S^\top e_i\|^2 \leq \mu r/n, \ \max_i \|D^\top e_i\|^2 \leq \mu r/n, \ \|SD^\top\|_\infty^2 \leq \mu r/n^2. \tag{7}$$

**Theorem 1** *Suppose that $X_0$ satisfies the incoherence condition given by Eq. (7), and $\Omega$ is uniformly distributed among all sets of size $\omega \geq n^2/10$. We assume that each entry is corrupted independently with probability $q$. Let $X$ and $Y$ be the solution of optimization problem given by Eq. (2) with parameter $\tau = O(1/\sqrt{n})$, $\kappa = O(1/\sqrt{n})$ and $\alpha = O(1/n)$. For some constant $c > 0$ and sufficiently large $n$, the following holds with probability at least $1 - cn^{-10}$,*

$$\|X_0 - X\|_F \leq \delta \text{ and } \|Y_0 - Y\|_F \leq \delta$$

*if $\mathrm{rank}(X_0) \leq \rho_r n/\mu/log^2 n$ and $q \leq q_s$, where $\rho_r$ and $q_s$ are positive constants.*

We now prove the global convergence of UMA with a worst-case $O(1/t)$ convergence rate measured by iteration complexity. Let $U = (Z; X; Y)$ and $W = (Z; X; Y; \Lambda)$. We also define

$$\theta(U) = \|X\|_* + \tau\|Y\|_1 - \alpha\langle M, Y\rangle + \frac{\kappa}{2}\|Y\|_F^2 \text{ and } U_t^{k+1} = \frac{1}{t}\sum_{k=1}^t U^{k+1}.$$

It follows from Corollaries 28.2.2 and 28.3.1 of [29] that the solution set of Eq. (2) is non-empty. Then, let $W^* = ((Z^*)^\top, (X^*)^\top, (Y^*)^\top, (\Lambda^*)^\top)^\top$ be a saddle point of Eq. (2), and define $U^* = ((Z^*)^\top, (X^*)^\top, (Y^*)^\top)^\top$.

**Theorem 2** *For $t$ iterations generated by UMA with $\beta \in \left(0, (\sqrt{33} - 5)\kappa/2\right)$,*

1) *We have $\|X_t^{k+1} + Y_t^{k+1} + Z_t^{k+1} - P_\Omega M\|^2 \leq \bar{c}_1/t^2$ for some constant $\bar{c}_1 > 0$.*
2) *We have $|\theta(U_t^{k+1}) - \theta(U^*)| \leq \bar{c}_2/t$ for some constant $\bar{c}_2 > 0$.*

## 6 Experiments

In this section, we compare our proposed UMA with the state-of-the-art approaches for attack detection. We consider three common evaluating metrics for attack detection as in [13]:

$$\text{Precision} = \frac{\text{TP}}{\text{TP} + \text{FP}}, \ \text{Recall} = \frac{\text{TP}}{\text{TP} + \text{FN}}, \ \text{F1} = \frac{2 \times \text{Precision} \times \text{Recall}}{\text{Precision} + \text{Recall}}$$

where TP is the number of attack profiles correctly detected as attacks, FP is the number of normal profiles that are misclassified, and FN is the number of attack profiles that are misclassified.

### 6.1 Datasets

We first conduct our experiments on the common-used datasets MovieLens100K and MovieLens1M, released by GroupLens [25]. These datasets are collected from a non-commercial recommender system, and it is more likely that the users in this dataset are non-spam users. We take the users already in the datasets as normal users. The rating scores range from 1 to 5, and we preprocess the data by minus 3 to the range $[-2, 2]$. Dataset MovieLens100K contains 100000 ratings of 943 users over 1682 movies, and dataset MovieLens1M contains 1000209 ratings of 6040 users over 3706 movies. We describe how to add attack profiles in Section 6.3.

We also collect a real dataset Douban10K[1] with attack profiles from Douban website, where registered users record rating information over various films, books, clothes, etc. We gather 12095 ratings of 213 users over 155 items. The rating scores range from 1 to 5, and we preprocess the data by minus 3 to the range $[-2, 2]$. Among the 213 user profiles, 35 profiles are attack profiles.

Table 1: Detection precision, recall and F1 on MovieLens100K and MovieLens1M. Here unorganized malicious attacks are based on a combination of traditional strategies.

| | MovieLens100K | | | MovieLens1M | | |
|---|---|---|---|---|---|---|
| | Precision | Recall | F1 | Precision | Recall | F1 |
| UMA | **0.934±0.003** | **0.883±0.019** | **0.908±0.011** | **0.739±0.009** | **0.785±0.023** | **0.761±0.016** |
| RPCA | 0.908±0.010 | 0.422±0.048 | 0.575±0.047 | 0.342±0.003 | 0.558±0.028 | 0.424±0.009 |
| N-P | 0.774±0.015 | 0.641±0.046 | 0.701±0.032 | 0.711±0.007 | 0.478±0.018 | 0.572±0.014 |
| $k$-means | 0.723±0.171 | 0.224±0.067 | 0.341±0.092 | 0.000±0.000 | 0.000±0.000 | 0.000±0.000 |
| PCAVarSel | 0.774±0.009 | 0.587±0.024 | 0.668±0.019 | 0.278±0.007 | 0.622±0.022 | 0.384±0.011 |
| MF-based | 0.911±0.009 | 0.814±0.008 | 0.860±0.009 | 0.407±0.005 | 0.365±0.004 | 0.385±0.005 |

Table 2: Detection precision, recall and F1 on MovieLens100K and MovieLens1M. Here unorganized malicious attacks consider the hire of existing users in addition to combination.

| | MovieLens100K | | | MovieLens1M | | |
|---|---|---|---|---|---|---|
| | Precision | Recall | F1 | Precision | Recall | F1 |
| UMA | **0.929±0.013** | **0.865±0.032** | **0.896±0.022** | **0.857±0.005** | **0.733±0.003** | **0.790±0.002** |
| RPCA | 0.797±0.046 | 0.659±0.097 | 0.721±0.097 | 0.635±0.012 | 0.391±0.022 | 0.484±0.015 |
| N-P | 0.244±0.124 | 0.145±0.089 | 0.172±0.084 | 0.273±0.020 | 0.099±0.031 | 0.144±0.035 |
| $k$-means | 0.767±0.029 | 0.234±0.042 | 0.357±0.051 | 0.396±0.026 | 0.300±0.039 | 0.341±0.035 |
| PCAVarSel | 0.481±0.027 | 0.168±0.017 | 0.248±0.023 | 0.120±0.006 | 0.225±0.012 | 0.157±0.008 |
| MF-based | 0.556±0.023 | 0.496±0.021 | 0.524±0.022 | 0.294±0.012 | 0.264±0.010 | 0.278±0.011 |

## 6.2 Comparison Methods and Implementation Details

We compare UMA with the state-of-the-art approaches for attack detection and robust PCA:

- **N-P:** A statistical algorithm based on the Neyman-Pearson statistics [16].
- **$k$-means:** A cluster algorithm based on classification attributes [3].
- **PCAVarSel:** A PCA-based variable selection algorithm [22].
- **MF-based:** A reputation estimation algorithm based on low-rank matrix factorization [19].
- **RPCA:** A low-rank matrix recovery method by considering sparse noise [9].

In the experiments, we set $\tau = 10/\sqrt{m}$, $\alpha = 10/m$ and $\delta = \sqrt{mn/200}$. A rating can be viewed as a malicious rating if it deviates from the ground-truth rating by more than 3, since the scale of ratings is from -2 to 2. We set parameter $\beta = \tau/3$ according to Eq. (6) where the entries of $Y$ will be nullified if they are smaller than the threshold. We set $\kappa = \tau$ under the convergence condition $\beta \in (0, (\sqrt{33} - 5)\kappa/2)$ as in Theorem 2. For the baseline methods, we take the results reported in [26] for comparison.

## 6.3 Comparison Results

In the first experiment, we add attack profiles into the datasets MovieLens100K and MovieLens1M by a combination of several traditional attack strategies. These traditional attack strategies include average attack strategy, random attack strategy and bandwagon attack strategy, discussed in Section 3.2. Specifically, each attacker randomly chooses one strategy to produce the user rating profiles and promotes one item randomly selected from items with average rating lower than 0. In line with the setting of previous attack detection works, we set the filler ratio (percentage of rated items in total items) as 0.01 and the filler items are drawn from the top 10% most popular items. We set the spam ratio (number of attack profiles/number of all user profiles) as 0.2. The experiment is repeated 10 times, and the average performance is reported. Table 1 shows the experimental results on datasets MovieLens100K and MovieLens1M under the attack profiles of a combination of traditional strategies.

The second experiment studies a more general case of unorganized malicious attacks. We consider that attackers can hire existing users to attack their targets, in addition to the profile injection attacks as mentioned above. We set spam ratio as 0.2, where 25% of the attack profiles are produced similar to the first experiment, and 75% of the attack profiles are from existing users by randomly changing the rating of one item lower than 0 to +2. In this case, attacks are more difficult to be detected, because the attack profiles are more similar to normal user profiles. The experiment is repeated 10 times and Table 2 demonstrates the comparison results on MovieLens100K and MovieLens1M.

Table 3: Detection precision, recall and F1 on dataset Douban10K.

| Methods | UMA | RPCA | N-P | $k$-means | PCAVarSel | MF-based |
|---|---|---|---|---|---|---|
| Precision | **0.800** | 0.535 | 0.250 | 0.321 | 0.240 | 0.767 |
| Recall | **0.914** | 0.472 | 0.200 | 0.514 | 0.343 | 0.657 |
| F1 | **0.853** | 0.502 | 0.222 | 0.396 | 0.282 | 0.708 |

Table 3 shows the experiments on dataset Douban10K. The experimental results in Table 1, 2 and 3 show that our proposed algorithm UMA achieves the best performance on all the datasets and three measures: Precision, Recall and F1.

Traditional attack detection approaches perform ineffectively on unorganized malicious attacks detection, because the success of those methods depends on the properties of shilling attacks, e.g., $k$-means method and N-P method work well if the attack profiles are similar in the view of classification attributes or latent categories, and PCAVarSel method achieves good performance only if attack profiles have more common unrated items than normal profiles. In summary, these methods detect attacks by identifying some common characteristics of attack profiles, while these do not hold in unorganized malicious attacks. The RPCA and MF-based methods try to find the ground-truth rating matrix from the observed rating matrix, whereas they hardly separate the sparse attack matrix from the noisy matrix and tend to suffer from low precision, especially on large-scale and heavily sparse dataset MovieLens1M.

We compare UMA with other approaches by varying the spam ratio from 2% to 20% since different systems may contain different spam ratios (# attack profiles/# all user profiles). As can be shown in Figure 2, UMA is robust and achieves the best performance in different spam ratios, whereas the comparison methods (except the RPCA method) achieve worse performance for small spam ratio, e.g., the N-P approach detects almost nothing. Although the RPCA method is as stable as UMA in different spam ratios, there is a performance gap between RPCA and UMA which becomes bigger when the dataset gets larger and sparser from MovieLens100K to MovieLens1M.

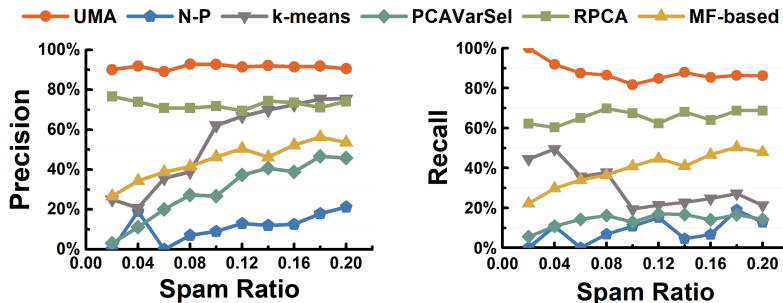

Figure 2: Detection precision and recall on MovieLens100K under unorganized malicious attacks. The spam ratio (# attack profiles/# all user profiles) varies from 0.02 to 0.2.

# 7   Conclusion

Attack detection plays an important role to improve the quality of recommendation. Most previous methods focus on shilling attacks, and the key idea for detecting such attacks is to find the common characteristics of attack profiles with the same attack strategy. This paper considers the unorganized malicious attacks, produced by multiple attack strategies to attack different targets. We formulate unorganized malicious attacks detection as a variant of matrix completion problem, and we propose the UMA algorithm and prove its recovery guarantee and global convergence. Experiments show that UMA achieves significantly better performance than the state-of-the-art methods for attack detection.

**Acknowledgments** This research was supported by the National Key R&D Program of China (2018YFB1004300), NSFC (61333014, 61503179), JiangsuSF (BK20150586), and Collaborative Innovation Center of Novel Software Technology and Industrialization, and Fundamental Research Funds for the Central Universities.

## Footnotes

[1] http://www.douban.com/.

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
