[Supplementary Material]

# Supplementary Material: Unorganized Malicious Attacks Detection

**Ming Pang**    **Wei Gao**    **Min Tao**    **Zhi-Hua Zhou**
National Key Laboratory for Novel Software Technology,
Nanjing University, Nanjing, 210023, China
{pangm, gaow, zhouzh}@lamda.nju.edu.cn   taom@nju.edu.cn

## 1  Optimization Model of UMA

We consider the following optimization model:

$$
\begin{aligned}
\min \quad & \|X\|_* + \tau\|Y\|_1 - \alpha\langle \bar{M}, Y\rangle + \frac{\kappa}{2}\|Y\|_F^2, \\
s.t. \quad & X + Y + Z = \bar{M}, \\
& Z \in \mathbf{B}, \\
& \mathbf{B} := \{Z\,|\,\|P_\Omega(Z)\|_F \le \delta\},
\end{aligned}
\tag{1.1}
$$

where $\kappa > 0$ is a regularization parameter and $\bar{M} := P_\Omega(M)$. The model (1.1) is a three-block convex programming. We define the Lagrangian function and augmented Lagrangian function of (1.1) as follows:

$$
\mathcal{L}(X, Y, Z, \Lambda) := \|X\|_* + \tau\|Y\|_1 - \alpha\langle \bar{M}, Y\rangle + \frac{\kappa}{2}\|Y\|_F^2 - \langle\Lambda, X + Y + Z - \bar{M}\rangle, \tag{1.2}
$$

$$
\mathcal{L}_\mathcal{A}(X, Y, Z, \Lambda, \beta) := \|X\|_* + \tau\|Y\|_1 - \alpha\langle \bar{M}, Y\rangle + \frac{\kappa}{2}\|Y\|_F^2 - \langle\Lambda, X + Y + Z - \bar{M}\rangle
$$
$$
+ \frac{\beta}{2}\|X + Y + Z - \bar{M}\|_F^2, \tag{1.3}
$$

where $\beta > 0$ is the penalty parameter.

## 2  Recovery Guarantee

In this section, we present theoretical guarantee that UMA can recover the low-rank component $X_0$ and the sparse component $Y_0$. For simplicity, our theoretical analysis focuses on square matrix, and it is natural to generalize our results to the general rectangular matrices.

Let the singular value decomposition of $X_0 \in \mathcal{R}^{n\times n}$ be given by

$$
X_0 = S\Sigma D^\top = \sum_{i=1}^r \sigma_i s_i d_i^\top \tag{2.4}
$$

where $r$ is the rank of matrix $X_0$, $\sigma_1, \ldots, \sigma_r$ are the positive singular values, and $S = [s_1, \ldots, s_r]$ and $D = [d_1, \ldots, d_r]$ are the left- and right-singular matrices, respectively. For $\mu > 0$, we assume

$$
\begin{aligned}
\max_i \|S^\top e_i\|^2 &\le \mu r/n, \\
\max_i \|D^\top e_i\|^2 &\le \mu r/n, \\
\|SD^\top\|_\infty^2 &\le \mu r/n^2.
\end{aligned}
\tag{2.5}
$$

Firstly, we consider the following optimization problem where all the entries of $M$ can be observed.

$$
\begin{aligned}
\min_{X,Y,Z} & \|X\|_* + \tau\|Y\|_1 - \alpha\langle M, Y\rangle + \frac{\kappa}{2}\|Y\|_F^2 \\
\text{s.t. } & X + Y + Z = M, \\
& \|Z\|_F \le \delta.
\end{aligned}
\tag{2.6}
$$

**Theorem 2.1** *Suppose that the support set of $Y_0$ be uniformly distributed for all sets of cardinality $k$, and $X_0$ satisfies the incoherence condition given by Eqn. (2.5). Let $X$ and $Y$ be the solution of optimization problem given by Eqn. (2.6) with parameter $\tau = O(1/\sqrt{n})$ , $\kappa = O(1/\sqrt{n})$ and $\alpha = O(1/n)$. For some constant $c > 0$ and sufficiently large $n$, the following holds with probability at least $1 - cn^{-10}$ over the choice on the support of $Y_0$*

$$\|X_0 - X\|_F \leq \delta \text{ and } \|Y_0 - Y\|_F \leq \delta \tag{2.7}$$

*if $rank(X_0) \leq \rho_r n/\mu/log^2 n$ and $k \leq \rho_s n^2$, where $\rho_r$ and $\rho_s$ are positive constant.*

*Proof:*

Let $\Omega$ be the space of matrices with the same support as $Y_0$, and let $T$ denote the linear space of matrices

$$T := \{SA^\top + BD^\top, A, B \in \mathbb{R}^{n \times r}\}. \tag{2.8}$$

We will first prove that, for $\|P_\Omega P_T\| \leq 1/2$, $(X_0, Y_0)$ is the unique solution if there is a pair $(W, F)$ satisfying

$$SD^\top + W = \tau(\text{sgn}(Y_0) + F + P_\Omega K) \tag{2.9}$$

where $P_T W = 0$ and $\|W\| \leq 1/2$, $P_\Omega F = 0$ and $\|F\|_\infty \leq 1/2$ and $\|P_\Omega K\|_F \leq 1/4$. Notice that $SD^\top + W_0$ is an arbitrary subgradient of $\|X\|_*$ at $(X_0, Y_0)$, and $\tau(\text{sgn}(Y_0) + F_0) - \alpha M + \kappa Y_0$ is an arbitrary subgradient of $\tau\|Y\|_1 - \alpha\langle M, Y\rangle + \kappa\|Y\|_F^2/2$ at $(X_0, Y_0)$. For any matrix $H$, we have, by the definition of subgradient,

$$\|X_0 + H\|_* + \tau\|Y_0 - H\|_1 - \alpha\langle M, Y_0 - H\rangle + \frac{\kappa}{2}\|Y_0 - H\|_F^2$$
$$\geq \|X_0\|_* + \tau\|Y_0\|_1 - \alpha\langle M, Y_0\rangle + \frac{\kappa}{2}\|Y_0\|_F^2 + \langle\alpha M - \kappa Y_0, H\rangle$$
$$+ \langle SD^\top + W_0, H\rangle - \tau\langle\text{sgn}(Y_0) + F_0, H\rangle. \tag{2.10}$$

By setting $W_0$ and $F_0$ satisfying $\langle W_0, H\rangle = \|P_{T^\perp}H\|_*$ and $\langle F_0, H\rangle = -\|P_{\Omega^\perp}H\|_1$, we have

$$\langle SD^\top + W_0, H\rangle - \tau\langle\text{sgn}(Y_0) + F_0, H\rangle$$
$$= \|P_{T^\perp}H\|_* + \tau\|P_{\Omega^\perp}H\|_1 + \langle SD^\top - \tau\text{sgn}(Y_0), H\rangle$$
$$= \|P_{T^\perp}H\|_* + \tau\|P_{\Omega^\perp}H\|_1 + \langle\tau(F + P_\Omega K) - W, H\rangle$$
$$\geq \frac{1}{2}(\|P_{T^\perp}H\|_* + \tau\|P_{\Omega^\perp}H\|_1) + \tau\langle P_\Omega K, H\rangle \tag{2.11}$$

where the second equality holds from Eqn. (2.9), and the last inequality holds from

$$\langle\tau F - W, H\rangle \geq -|\langle W, H\rangle| - |\langle\tau F, H\rangle| \geq -(\|P_{T^\perp}H\|_* + \tau\|P_{\Omega^\perp}H\|_1)/2$$

for $\|W\| \leq 1/2$ and $\|F\|_\infty \leq 1/2$. We further have

$$\langle\tau P_\Omega K, H\rangle \geq -\frac{\tau}{4}\|P_{\Omega^\perp}H\|_F - \frac{\tau}{2}\|P_{T^\perp}H\|_F \tag{2.12}$$

from $\|P_\Omega K\|_F \leq 1/4$ and

$$\|P_\Omega H\|_F \leq \|P_\Omega P_T H\|_F + \|P_\Omega P_{T^\perp}H\|_F \leq \|P_\Omega P_{T^\perp}H\|_F + \|H\|_F/2$$
$$\leq (\|P_\Omega H\|_F + \|P_{\Omega^\perp}H\|_F)/2 + \|P_\Omega P_{T^\perp}H\|_F.$$

Combining with Eqns. (2.10) to (2.12), we have

$$\|X_0 + H\|_* + \tau\|Y_0 - H\|_1 - \alpha\langle M, Y_0 - H\rangle + \tfrac{\kappa}{2}\|Y_0 - H\|_F^2$$
$$\geq \quad \|X_0\|_* + \tau\|Y_0\|_1 - \alpha\langle M, Y_0\rangle + \tfrac{\kappa}{2}\|Y_0\|_F^2$$
$$+\langle \alpha M - \kappa Y_0, H\rangle + \tfrac{1-\tau}{2}\|P_{T^\perp}H\|_* + \tfrac{\tau}{4}\|P_{\Omega^\perp}H\|_1$$

From the conditions that $\Omega \cap T = \{0\}$, $\tau = O(1/\sqrt{n})$, $\kappa = O(1/\sqrt{n})$ and $\alpha = O(1/n)$, we have

$$\langle \alpha M - \kappa Y_0, H\rangle + \frac{1-\tau}{2}\|P_{T^\perp}H\|_* + \frac{\tau}{4}\|P_{\Omega^\perp}H\|_1 > 0 \tag{2.13}$$

for sufficient large $n$. Therefore, we can recover $X_0$ and $Y_0$ if there is a pair $(W, F)$ satisfying Eqn. (2.9), and the pair $(W, F)$ can be easily constructed according to [7]. We complete the proof from the condition $\|Z\|_F \leq \delta$.

Similarly to the proof of Theorem 2.1, we present the following theorem for the minimization problem of Eqn. (1.1).

**Theorem 2.2** *Suppose that $X_0$ satisfies the incoherence condition given by Eqn. (2.5), and $\Omega$ is uniformly distributed among all sets of size $m \geq n^2/10$. We assume that each entry is corrupted independently with probability $q$. Let $X$ and $Y$ be the solution of optimization problem given by Eqn. (1.1) with parameter $\tau = O(1/\sqrt{n})$, $\kappa = O(1/\sqrt{n})$ and $\alpha = O(1/n)$. For some constant $c > 0$ and sufficiently large $n$, the following holds with probability at least $1 - cn^{-10}$*

$$\|X_0 - X\|_F \leq \delta \text{ and } \|Y_0 - Y\|_F \leq \delta \tag{2.14}$$

*if rank($X_0$)$\leq \rho_r n/\mu/log^2 n$ and $q \leq q_s$, where $\rho_r$ and $q_s$ are positive constants.*

## 3 Optimality condition

Before starting to show the convergence, we derive its optimality condition of (1.1). Let $\mathcal{W} := \mathbf{B} \times \mathcal{R}^{m\times n} \times \mathcal{R}^{m\times n} \times \mathcal{R}^{m\times n}$. It follows from Corollaries 28.2.2 and 28.3.1 of [1] that the solution set of (1.1) is non-empty. Then, let $W^* = ((Z^*)^\top, (X^*)^\top, (Y^*)^\top, (\Lambda^*)^\top)^\top$ be a saddle point of (1.1). It is easy to see that (1.1) is equivalent to finding $W^* \in \mathcal{W}$ such that

$$\begin{cases} \langle Z - Z^*, -\Lambda^*\rangle \geq 0, \\ \|X\|_* - \|X^*\|_* + \langle X - X^*, -\Lambda^*\rangle \geq 0, \\ \tau\|Y\|_1 - \tau\|Y^*\|_1 + \langle Y - Y^*, -\alpha\bar{M} + \kappa Y^* - \Lambda^*\rangle \geq 0, \\ X^* + Y^* + Z^* - \bar{M} = 0, \end{cases} \quad \forall W = \begin{pmatrix} Z \\ X \\ Y \\ \Lambda \end{pmatrix} \in \mathcal{W}, \tag{3.15}$$

or, in a more compact form:

$$\text{VI}(\mathcal{W}, \Psi, \theta) \qquad \theta(U) - \theta(U^*) + \langle W - W^*, \Psi(W^*)\rangle \geq \frac{\kappa}{2}\|Y - Y^*\|_F^2, \quad \forall W \in \mathcal{W}, \tag{3.16a}$$

where

$$U = \begin{pmatrix} Z \\ X \\ Y \end{pmatrix}, \quad \theta(U) = \|X\|_* + \tau\|Y\|_1 - \alpha\langle\bar{M}, Y\rangle + \frac{\kappa}{2}\|Y\|_F^2, \tag{3.16b}$$

$$\text{and } W = \begin{pmatrix} Z \\ X \\ Y \\ \Lambda \end{pmatrix}, \quad V = \begin{pmatrix} X \\ Y \\ \Lambda \end{pmatrix}, \quad \Psi(W) = \begin{pmatrix} -\Lambda \\ -\Lambda \\ -\Lambda \\ X + Y + Z - \bar{M} \end{pmatrix}. \tag{3.16c}$$

Note that $U$ collects all the primal variables in (3.15) and it is a sub-vector of $W$. Moreover, we use $\mathcal{W}^*$ to denote the solution set of $\text{VI}(\mathcal{W}, \Psi, \theta)$ and define $V^* = ((X^*)^\top, (Y^*)^\top, (\Lambda^*)^\top)^\top$ and $\mathcal{V}^* := \{V^*|W^* \in \mathcal{W}^*\}$.

# 4 Convergence Analysis

In this section, we solve (1.1) with global convergence. More specifically, let $(X^k, Y^k, \Lambda^k)$ be given, UMA generates the new iterate $W^{k+1}$ via the following scheme:

$$
\begin{cases}
Z^{k+1} = \arg\min_{Z \in \mathbf{B}} \mathcal{L}_A(X^k, Y^k, Z, \Lambda^k, \beta), \\
X^{k+1} = \arg\min_{X \in \mathcal{R}^{m \times n}} \mathcal{L}_A(X, Y^k, Z^{k+1}, \Lambda^k, \beta), \\
Y^{k+1} = \arg\min_{Y \in \mathcal{R}^{m \times n}} \mathcal{L}_A(X^{k+1}, Y, Z^{k+1}, \Lambda^k, \beta), \\
\Lambda^{k+1} = \Lambda^k - \beta(X^{k+1} + Y^{k+1} + Z^{k+1} - \bar{M}),
\end{cases}
\tag{4.1}
$$

which can be easily written into the following more specific form:

$$
Z^{k+1} = \arg\min_{Z \in \mathbf{B}} \frac{\beta}{2} \|Z + X^k + Y^k - \frac{1}{\beta}\Lambda^k - \bar{M}\|_F^2, \tag{4.2}
$$

$$
X^{k+1} = \arg\min_{X \in \mathcal{R}^{m \times n}} \|X\|_* + \frac{\beta}{2} \|X + Y^k + Z^{k+1} - \frac{1}{\beta}\Lambda^k - \bar{M}\|_F^2, \tag{4.3}
$$

$$
Y^{k+1} = \arg\min_{Y \in \mathcal{R}^{m \times n}} \tau \|Y\|_1 - \alpha\langle \bar{M}, Y\rangle + \frac{\kappa}{2}\|Y\|_F^2
$$
$$
+ \frac{\beta}{2}\|Y + X^{k+1} + Z^{k+1} - \frac{1}{\beta}\Lambda^k - \bar{M}\|_F^2, \tag{4.4}
$$

$$
\Lambda^{k+1} = \Lambda^k - \beta(X^{k+1} + Y^{k+1} + Z^{k+1} - \bar{M}). \tag{4.5}
$$

In the following, we concentrate on the convergence of UMA. In contrast to the existing results in [6], we aim to present a much more sharp result. We first prove some properties of the sequence generated by UMA, which play a crucial role in the coming convergence analysis. Before that, we introduce some notations:

$$
\Delta_\Lambda := \frac{1}{2\beta}(\|\Lambda^{k+1} - \Lambda\|_F^2 - \|\Lambda^k - \Lambda\|_F^2 + \|\Lambda^{k+1} - \Lambda^k\|_F^2), \tag{4.6}
$$

$$
\Delta_X := \frac{1}{2\beta}(\|X^{k+1} - X\|_F^2 - \|X^k - X\|_F^2 + \|X^{k+1} - X^k\|_F^2), \tag{4.7}
$$

$$
\Delta_Y := \frac{1}{2\beta}(\|Y^{k+1} - Y\|_F^2 - \|Y^k - Y\|_F^2 + \|Y^{k+1} - Y^k\|_F^2), \tag{4.8}
$$

$$
\mathcal{R} = X + Y + Z - \bar{M}, \tag{4.9}
$$

$$
\mathcal{R}^{k+1} = X^{k+1} + Y^{k+1} + Z^{k+1} - \bar{M}. \tag{4.10}
$$

**Lemma 4.1** *Let $\{W^k\}$ be generated by UMA. Then, we have*

*(1)*

$$
\langle \Lambda^k - \Lambda^{k+1}, Y^k - Y^{k+1}\rangle \geq \kappa\|Y^k - Y^{k+1}\|_F^2. \tag{4.11}
$$

*(2)*

$$
\langle \Lambda^k - \Lambda^{k+1}, X^k - X^{k+1}\rangle \geq \beta\langle X^k - X^{k+1}, Y^{k+1} - Y^k - (Y^k - Y^{k-1})\rangle \tag{4.12}
$$

*Proof:* (1) Using the optimality of (4.4), we get

$$
\langle Y - Y^{k+1}, \partial(\tau\|Y^{k+1}\|_1) - \Lambda^{k+1} - \alpha\bar{M} + \kappa Y^{k+1}\rangle \geq 0. \tag{4.13}
$$

Setting $Y := Y^k$ in (4.13), we have

$$
\langle Y^k - Y^{k+1}, \partial(\tau\|Y^{k+1}\|_1) - \Lambda^{k+1} - \alpha\bar{M} + \kappa Y^{k+1}\rangle \geq 0. \tag{4.14}
$$

Then, setting $Y := Y^{k+1}$ in (4.13) with the index $k$ replaced with $k-1$, it yields

$$
\langle Y^{k+1} - Y^k, \partial(\tau\|Y^k\|_1) - \Lambda^k - \alpha\bar{M} + \kappa Y^k\rangle \geq 0. \tag{4.15}
$$

Thus, adding (4.14) and (4.15) together, the inequality (4.11) follows directly.

(2) The inequality (4.12) can be proved in a similar way as (4.11).

**Lemma 4.2** *Let $\{W^k\}$ be generated by UMA. Then, we have the following inequality:*

$$\theta(U) - \theta(U^{k+1}) + \langle W - W^{k+1}, \Psi(W^{k+1})\rangle + \beta\langle \mathcal{R}, \Gamma(X^k, Y^k, Z^k)\rangle$$

$$\geq \frac{1}{2}(\|V^{k+1} - V\|_Q^2 + \|V^k - V^{k+1}\|_Q^2 - \|V^k - V\|_Q^2) + \kappa\|Y^{k+1} - Y^k\|_F^2$$

$$+ \frac{\kappa}{2}\|Y^{k+1} - Y\|_F^2 - \beta\langle X^{k+1} - X^k, Y^{k+1} - Y^k - (Y^k - Y^{k-1})\rangle$$

$$+ \beta\langle Y^{k+1} - Y, X^{k+1} - X^k\rangle. \tag{4.16}$$

*where*

$$\Gamma(X^k, Y^k, Z^k) = Y^k - Y^{k+1} + X^k - X^{k+1},$$

$$Q = \begin{pmatrix} \beta I & 0 & 0 \\ 0 & \beta I & 0 \\ 0 & 0 & \frac{1}{\beta}I \end{pmatrix} \tag{4.17}$$

*Proof:* According to the optimality condition of (4.1), we have

$$\begin{cases} \langle Z - Z^{k+1}, -\Lambda^{k+1} + \beta(X^k - X^{k+1}) + \beta(Y^k - Y^{k+1})\rangle \geq 0, \\ \|X\|_* - \|X^{k+1}\|_* + \langle X - X^{k+1}, -\Lambda^{k+1} + \beta(Y^k - Y^{k+1})\rangle \geq 0, \\ \tau\|Y\|_1 - \tau\|Y^{k+1}\|_1 + \langle Y - Y^{k+1}, -\alpha\bar{M} + \kappa Y^{k+1} - \Lambda^{k+1}\rangle \geq 0, \\ \langle \Lambda - \Lambda^{k+1}, X^{k+1} + Y^{k+1} + Z^{k+1} - \bar{M} - \frac{1}{\beta}(\Lambda^k - \Lambda^{k+1})\rangle \geq 0, \end{cases} \quad \forall W = \begin{pmatrix} Z \\ X \\ Y \\ \Lambda \end{pmatrix} \tag{4.18}$$

Then, combining the above inequalities with (3.16b) and (3.16c), we get

$$\theta(U) - \theta(U^{k+1}) + \langle W - W^{k+1}, \Psi(W^{k+1})\rangle + \beta\left(\langle Z - Z^{k+1}, Y^k - Y^{k+1} + X^k - X^{k+1}\rangle\right)$$

$$+ \beta\langle X - X^{k+1}, Y^k - Y^{k+1}\rangle\big) \geq \frac{1}{2\beta}\Delta_\Lambda + \frac{\kappa}{2}\|Y - Y^{k+1}\|_F^2.$$

Then, invoking (4.9) and (4.10), we obtain that

$$\theta(U) - \theta(U^{k+1}) + \langle W - W^{k+1}, \Psi(W^{k+1})\rangle + \beta\langle \mathcal{R} - \mathcal{R}^{k+1}, Y^k - Y^{k+1} + X^k - X^{k+1}\rangle$$

$$\geq \frac{\kappa}{2}\|Y - Y^{k+1}\|_F^2 + \frac{1}{2\beta}\Delta_\Lambda + \frac{\beta}{2}(\Delta_X + \Delta_Y) + \beta\langle Y - Y^{k+1}, X^k - X^{k+1}\rangle.$$

Thus, using $\mathcal{R}^{k+1} = \frac{1}{\beta}(\Lambda^k - \Lambda^{k+1})$, it yields that

$$\theta(U) - \theta(U^{k+1}) + \langle W - W^{k+1}, \Psi(W^{k+1})\rangle + \beta\langle \mathcal{R}, Y^k - Y^{k+1} + X^k - X^{k+1}\rangle$$

$$\geq \frac{\kappa}{2}\|Y - Y^{k+1}\|_F^2 + \frac{1}{2\beta}\Delta_\Lambda + \frac{\beta}{2}(\Delta_X + \Delta_Y) + \beta\langle Y - Y^{k+1}, X^k - X^{k+1}\rangle$$

$$+ \langle \Lambda^k - \Lambda^{k+1}, Y^k - Y^{k+1} + X^k - X^{k+1}\rangle. \tag{4.19}$$

On the other hand, adding (4.11) and (4.12) together, we obtain that

$$\langle \Lambda^k - \Lambda^{k+1}, Y^k - Y^{k+1} + X^k - X^{k+1}\rangle$$

$$\geq \kappa\|Y^k - Y^{k+1}\|_F^2 - \beta\langle X^{k+1} - X^k, Y^{k+1} - Y^k - (Y^k - Y^{k-1})\rangle.$$

Next, substituting the above inequality into (4.19), and invoking, it yields the assertion (4.16).

$\square$

In the following, we give each crossing term in the right-hand of (4.16) a low bound. The following inequalities enable us to get a much sharper result for UMA solving (1.1) in contrast to ([6]).

**Lemma 4.3** *Let $\{W^k\}$ be generated by UMA. Suppose that $0 < \varepsilon < \sqrt{5} - 2$. Then, it holds that*

$$- \beta\langle X^{k+1} - X^k, Y^{k+1} - Y^k\rangle \geq \beta\left(-\frac{3 - \sqrt{5}}{4}\|X^k - X^{k+1}\|_F^2 - \frac{\|Y^{k+1} - Y^k\|_F^2}{3 - \sqrt{5}}\right), \tag{4.20}$$

$$\beta\langle X^{k+1} - X^k, (Y^k - Y^{k-1})\rangle \geq \beta\left(-\frac{3 - \sqrt{5}}{4}\|X^k - X^{k+1}\|_F^2 - \frac{\|Y^k - Y^{k-1}\|_F^2}{3 - \sqrt{5}}\right), \tag{4.21}$$

$$\beta\langle Y^{k+1} - Y, X^{k+1} - X^k\rangle \geq -\beta\left(\frac{\|Y^{k+1} - Y\|_F^2}{2(\sqrt{5} - 2 - \varepsilon)} + \frac{\sqrt{5} - 2 - \varepsilon}{2}\|X^{k+1} - X^k\|_F^2\right). \tag{4.22}$$

*Proof:* These three inequalities follow from Cauchy-Schwarz inequality. □

**Theorem 4.4** *Let $\{W^k\}$ be generated by UMA. Assume that $\beta > 0$ in Algorithm (4.1). Suppose that $0 < \varepsilon < \sqrt{5} - 2$. Then, we have the following contractive property:*

$$
\frac{\beta}{2}\|X^{k+1} - X^*\|_F^2 + \frac{\beta}{2}\|Y^{k+1} - Y^*\|_F^2 + \frac{1}{2\beta}\|Z^{k+1} - Z^*\|_F^2 + \frac{\beta}{3 - \sqrt{5}}\|Y^k - Y^{k+1}\|_F^2
$$

$$
\leq \frac{\beta}{2}\|X^k - X^*\|_F^2 + \frac{\beta}{2}\|Y^k - Y^*\|_F^2 + \frac{1}{2\beta}\|Z^k - Z^*\|_F^2 + \frac{\beta}{3 - \sqrt{5}}\|Y^{k-1} - Y^k\|_F^2
$$

$$
- \frac{\varepsilon}{2}\beta\|X^k - X^{k+1}\|_F^2 - (\kappa - \frac{\sqrt{5} + 2}{2}\beta)\|Y^{k+1} - Y^k\|_F^2 - \frac{1}{2\beta}\|\Lambda^k - \Lambda^{k+1}\|_F^2
$$

$$
- (\kappa - \frac{1}{2(\sqrt{5} - 2 - \varepsilon)\beta})\|Y^{k+1} - Y^*\|_F^2. \tag{4.23}
$$

*Proof:* First, invoking (3.16a) and $X^* + Y^* + Z^* - \bar{M} = 0$, we have

$$
\theta(U^{k+1}) - \theta(U^*) + \langle W^{k+1} - W^*, \Psi(W^{k+1})\rangle + \beta\langle X^* + Y^* + Z^* - \bar{M}, \Gamma(X^k, Y^k, Z^k)\rangle
$$

$$
\geq \frac{\kappa}{2}\|Y^{k+1} - Y^*\|_F^2. \tag{4.24}
$$

Then, setting $W := W^* \in \mathcal{W}^*$ in (4.16) and combining with (4.24), we obtain that

$$
0 \geq \frac{\beta}{2}(\|V^{k+1} - V^*\|_Q^2 + \|V^k - V^{k+1}\|_Q^2 - \|V^k - V^*\|_Q^2) + \kappa\|Y^{k+1} - Y^k\|_F^2 + \kappa\|Y^{k+1} - Y^*\|_F^2
$$

$$
- \beta\langle X^{k+1} - X^k, Y^{k+1} - Y^k - (Y^k - Y^{k-1})\rangle + \beta\langle Y^{k+1} - Y, X^{k+1} - X^k\rangle. \tag{4.25}
$$

Next, adding (4.20)-(4.22) together, then substituting the resulting inequality into (4.25), we derive the assertion (4.23) directly. □

Based on the above theorem, we have the following theorem immediately.

**Theorem 4.5** *When $\beta$ is restricted by*

$$
\beta \in \left(0, \ 2(\sqrt{5} - 2)\kappa\right), \tag{4.26}
$$

*there exists a sufficient small scalar $\varepsilon > 0$ such that*

$$
\kappa - \frac{\sqrt{5} + 2}{2}\beta > 0, \ \text{and} \ \kappa - \frac{1}{2(\sqrt{5} - 2 - \varepsilon)\beta} > 0. \tag{4.27}
$$

*Then, we have*

(1) *The sequence $\{V^k\}$ is bounded.*

(2) $\lim_{k\to\infty}\{\|Y^k - Y^{k+1}\|_F^2 + \|X^k - X^{k+1}\|_F^2 + \|\Lambda^k - \Lambda^{k+1}\|_F^2\} = 0.$

*Proof.* The inequality (4.27) is elementary. Note that the assertion (1) follows from (4.23) directly. Furthermore, we get

$$
\sum_{k=1}^{\infty}\left[\frac{\varepsilon}{2}\beta\|X^k - X^{k+1}\|_F^2 + (\kappa - \frac{\sqrt{5} + 2}{2}\beta)\|Y^{k+1} - Y^k\|_F^2 + \frac{1}{2\beta}\|\Lambda^k - \Lambda^{k+1}\|_F^2\right]
$$

$$
\leq \frac{\beta}{2}\|X^1 - X^*\|_F^2 + \frac{\beta}{2}\|Y^1 - Y^*\|_F^2 + \frac{1}{2\beta}\|Z^1 - Z^*\|_F^2 + \frac{\beta}{3 - \sqrt{5}}\|Y^0 - Y^1\|_F^2 < +\infty,
$$

which immediately implies that

$$
\lim_{k\to\infty}\|Y^k - Y^{k+1}\|_F = 0, \quad \lim_{k\to\infty}\|X^k - X^{k+1}\|_F = 0, \quad \lim_{k\to\infty}\|\Lambda^k - \Lambda^{k+1}\|_F = 0, \tag{4.28}
$$

i.e., the second assertion. □

We are now ready to prove the convergence of UMA.

**Theorem 4.6** *Let $\{V^k\}$ and $\{W^k\}$ be the sequences generated by UMA. Assume that the penalty parameter $\beta$ is satisfied with (4.26). Then, we have*

1. *Any cluster point of $\{W^k\}$ is a solution point of (3.15).*

2. *The sequence $\{V^k\}$ converges to some $V^\infty \in \mathcal{V}^*$.*

3. *The sequence $\{U^k\}$ converges to a solution point of (1.1).*

*Proof:* Since $\{W^k\}$ is bounded due to (4.23), it has at least one cluster point. Let $W^\infty$ be a cluster point of $\{W^k\}$ and the subsequence $\{W^{k_j}\}$ converges to $W^\infty$. Because of the assertion (4.28), it follows from (4.18) that

$$
\begin{cases}
\langle Z - Z^\infty, -\Lambda^\infty \rangle \geq 0, \\
\|X\|_* - \|X^\infty\|_* + \langle X - X^\infty, -\Lambda^\infty \rangle \geq 0, \\
\tau\|Y\|_1 - \tau\|Y^\infty\|_1 + \langle Y - Y^\infty, -\alpha\bar{M} + \kappa Y^\infty - \Lambda^\infty \rangle \geq 0, \\
\langle \Lambda - \Lambda^\infty, X^\infty + Y^\infty + Z^\infty - \bar{M} \rangle \geq 0,
\end{cases}
\quad \forall\, W = \begin{pmatrix} Z \\ X \\ Y \\ \Lambda \end{pmatrix} \in \mathcal{W},
$$

Thus,

$$
\theta(U) - \theta(U^\infty) + (W - W^\infty)^\top \Psi(W^\infty) \geq \frac{\kappa}{2}\|Y - Y^\infty\|_F^2, \quad \forall\, W = (Z^\top, X^\top, Y^\top, \Lambda^\top)^\top \in \mathcal{W}.
$$

This means that $W^\infty$ is a solution of $\mathrm{VI}(\mathcal{W}, \Psi, \theta)$. Then the inequality (4.23) is also valid if $V^*$ is replaced by $V^\infty$. Therefore, the non-increasing sequence $\{\frac{1}{2}\|V^k - V^\infty\|_Q^2 + \frac{\beta}{3-\sqrt{5}}\|Y^k - Y^{k+1}\|_F^2\}$ converges to 0 since it has a subsequence $\{\frac{1}{2}\|V^{k_j} - V^\infty\|_Q^2 + \frac{\beta}{3-\sqrt{5}}\|Y^{k_j} - Y^{k_j+1}\|_F^2\}$ converges to 0. Thus, the sequence $\{V^k\}$ converges to some $V^\infty \in \mathcal{V}^*$. Also, the updating scheme of $\Lambda^{k+1}$ in (4.1) implies that

$$
Z^{k+1} = \bar{M} - X^{k+1} - Y^{k+1} + \frac{1}{\beta}(\Lambda^k - \Lambda^{k+1}).
$$

Combining the above equality, (4.28) and $\lim_{k\to\infty}\|V^k - V^\infty\|_Q^2 = 0$, we have $W^k$ converges to $W^\infty$. It implies that the sequence $U^k$ converges to a solution point of (1.1). Thus, the third assertion holds.

**Remark 4.7** *Note that the range for $\beta$ in ([6]) with convergence guarantee is $(0, 0.4\kappa)$ for UMA solving (1.1). However, we get a much larger range for the penalty parameter $\beta$ in (4.26).*

Next, we present a worst-case $O(1/t)$ convergence rate measured by the iteration complexity for UMA. Indeed, the range of $\beta$ to ensure the $O(1/t)$ convergence rate is slightly more restrictive than (4.26). Let us define

$$
Z_t^{k+1} = \frac{1}{t}\sum_{k=1}^t Z^{k+1}, \quad X_t^{k+1} = \frac{1}{t}\sum_{k=1}^t X^{k+1}, \quad Y_t^{k+1} = \frac{1}{t}\sum_{k=1}^t Y^{k+1},
$$

and

$$
U_t^{k+1} = \frac{1}{t}\sum_{k=1}^t U^{k+1}, \quad W_t^{k+1} = \frac{1}{t}\sum_{k=1}^t W^{k+1}.
$$

Obviously, $W_t^{k+1} \in \mathcal{W}$ because of the convexity $\mathcal{W}$. By invoking Theorem 4.5, there exists a constant $C$ such that

$$
\max\left(\|X^k\|_F, \|Y^k\|_F, \|Z^k\|_F, \|\Lambda^k\|_F\right) \leq C, \quad \forall\, k.
$$

Next, we present several lemmas to facilitate the convergence rate analysis.

**Lemma 4.8** *Let $\{W^k\}$ be generated by UMA. Suppose that $0 < \varepsilon < \sqrt{33} - 5$. Then, it holds that*

$$- \beta \langle X^{k+1} - X^k, Y^{k+1} - Y^k \rangle \geq \beta \left( -\frac{7 - \sqrt{33}}{8} \|X^k - X^{k+1}\|_F^2 - \frac{7 + \sqrt{33}}{8} \|Y^{k+1} - Y^k\|_F^2 \right),$$
(4.29)

$$\beta \langle X^{k+1} - X^k, (Y^k - Y^{k-1}) \rangle \geq \beta \left( -\frac{7 - \sqrt{33}}{8} \|X^k - X^{k+1}\|_F^2 - \frac{7 + \sqrt{33}}{8} \|Y^k - Y^{k-1}\|_F^2 \right),$$
(4.30)

$$\beta \langle Y^{k+1} - Y, X^{k+1} - X^k \rangle \geq -\beta \left( \frac{\|Y^{k+1} - Y\|_F^2}{\sqrt{33} - 5 - \varepsilon} + \frac{\sqrt{33} - 5 - \varepsilon}{4} \|X^{k+1} - X^k\|_F^2 \right). \quad (4.31)$$

*Proof:* These three inequalities follow from Cauchy-Schwarz inequality. $\square$

**Lemma 4.9** *Let $\{W^k\}$ be the sequence generated by UMA (4.1). If $\beta$ is restricted by*

$$\beta \in \left( 0, \frac{\sqrt{33} - 5}{2} \kappa \right),$$
(4.32)

*then we have*

$$\Theta(V^{k+1}, V^k, V) \leq \Theta(V^k, V^{k-1}, V) + \Xi(W^{k+1}, W^k, W),$$
(4.33)

*where*

$$\Theta(V^{k+1}, V^k, V) := \frac{1}{2} \|V^{k+1} - V\|_Q^2 + \frac{7 + \sqrt{33}}{8} \beta \|Y^{k+1} - Y^k\|_F^2.$$
(4.34)

*and*

$$\Xi(W^{k+1}, W^k, W) := \theta(U) - \theta(U^{k+1}) + (W - W^{k+1})^\top \Psi(W)$$
$$+ \beta \langle \mathcal{R}, Y^k - Y^{k+1} + X^k - X^{k+1} \rangle. \quad (4.35)$$

*Proof:* First, summing inequalities (4.29)-(4.31) together, we get

$$\beta \langle X^{k+1} - X^k, Y^{k+1} - Y^k - (Y^k - Y^{k-1}) \rangle + \beta \langle Y^{k+1} - Y, X^{k+1} - X^k \rangle$$
$$\geq \quad \frac{\varepsilon - 2}{4} \beta \|X^{k+1} - X^k\|_F^2 - \frac{7 + \sqrt{33}}{8} \beta \|Y^{k+1} - Y^k\|_F^2$$
$$- \frac{7 + \sqrt{33}}{8} \beta \|Y^k - Y^{k-1}\|_F^2 - \frac{1}{\sqrt{33} - 5 - \varepsilon} \beta \|Y^{k+1} - Y\|_F^2.$$

Then, substituting the above inequality into (4.16) and invoking (4.34), (4.35), we obtain

$$\Theta(V^{k+1}, V^k, V) \quad \leq \quad \Theta(V^k, V^{k-1}, V) + \Xi(W^{k+1}, W^k, W) - X^k\|_F^2$$
$$- (\kappa - \frac{5 + \sqrt{33}}{4} \beta) \|Y^{k+1} - Y^k\|_F^2 - \frac{\beta}{4} \varepsilon \|X^{k+1} - \frac{1}{2\beta} \|\Lambda^k - \Lambda^{k+1}\|_F^2$$
$$- (\frac{\kappa}{2} - \frac{1}{\sqrt{33} - 5 - \varepsilon} \beta) \|Y^{k+1} - Y\|_F^2.$$

Let $\varepsilon \to 0+$, the assertion follows directly.

**Theorem 4.10** *For $t$ iterations generated by UMA with $\beta$ restricted in (4.32), the following assertions holds.*

*(1) We have*

$$\theta(U_t^{k+1}) - \theta(U) + (W_t^{k+1} - W)^\top \Psi(W)$$
$$\leq \frac{1}{t} \left[ 4\beta C \|X + Y + Z - \bar{M}\|_F + \frac{1}{2} \|V^1 - V\|_Q^2 + \frac{7 + \sqrt{33}}{8} \beta \|Y^1 - Y^0\|_F^2 \right]. \quad (4.36)$$

*(2) There exists a constant $\bar{c}_1 > 0$ such that*

$$\|X_t^{k+1} + Y_t^{k+1} + Z_t^{k+1} - \bar{M}\|^2 \leq \frac{\bar{c}_1}{t^2}. \tag{4.37}$$

*(3) There exists a constant $\bar{c}_2 > 0$ such that*

$$|\theta(U_t^{k+1}) - \theta(U^*)| \leq \frac{\bar{c}_2}{t}. \tag{4.38}$$

*Proof:* 1) First, it follows from the assertion (4.33) that for all $W \in \mathcal{W}$, we have

$$\theta(U) - \theta(U^{k+1}) + (W - W^{k+1})^\top \Psi(W) + \beta\langle \mathcal{R}, Y^k - Y^{k+1} + X^k - X^{k+1}\rangle$$
$$\geq \Theta(V^{k+1}, V^k, V) - \Theta(V^k, V^{k-1}, V). \tag{4.39}$$

Summarizing both sides of the above inequalities from $k = 1, 2, \cdots, t$, we have

$$t\theta(U) - \sum_{k=1}^{t} \theta(U^{k+1}) + (tW - \sum_{k=1}^{t} W^{k+1})^\top \Psi(W) + \beta\langle \mathcal{R}, Y^1 - Y^{t+1} + X^1 - X^{t+1}\rangle$$
$$\geq \Theta(V^{t+1}, V^t, V) - \Theta(V^1, V^0, V). \tag{4.40}$$

Then, it follows from the convexity of $\theta$ that

$$\theta(U_t^{k+1}) \leq \frac{1}{t}\sum_{k=1}^{t} \theta(U^{k+1}). \tag{4.41}$$

Combining (4.40) and (4.41), we have

$$\theta(U_t^{k+1}) - \theta(U) + (W_t^{k+1} - W)^\top \Psi(W) \leq \frac{1}{t}\left(\Theta(V^1, V^0, V) + 4\beta C\|\mathcal{R}\|_F\right). \tag{4.42}$$

Thus, the assertion (4.36) follows from the above inequality and the defintion of $\Theta(V^1, V^0, V)$ directly.

2) Let us define $\bar{c}_1 = \frac{2}{\beta^2}\left(\|\Lambda^1 - \Lambda^*\|^2 + \|\Lambda^{k+1} - \Lambda^*\|^2\right)$. Then, we have

$$\|X_t^{k+1} + Y_t^{k+1} + Z_t^{k+1} - \bar{M}\|^2$$
$$= \left\|\frac{1}{t}\sum_{k=1}^{t}\left[X^{k+1} + Y^{k+1} + Z^{k+1} - \bar{M}\right]\right\|^2$$
$$= \left\|\frac{1}{t}\sum_{k=1}^{t}\left[\frac{1}{\beta}(\Lambda^k - \Lambda^{k+1})\right]\right\|^2 = \left\|\frac{1}{t\beta}\left(\Lambda^1 - \Lambda^{t+1}\right)\right\|^2$$
$$\leq \frac{2}{t^2\beta^2}\left(\|\Lambda^1 - \Lambda^*\|^2 + \|\Lambda^{k+1} - \Lambda^*\|^2\right) = \frac{\bar{c}_1}{t^2},$$

The assertion (4.37) is proved immediately.

3) It follows from $\mathcal{L}(U_t^{k+1}, \Lambda^*) \geq \mathcal{L}(U^*, \Lambda^*)$ with $\mathcal{L}$ defined in (1.2) that

$$\theta(U_t^{k+1}) - \theta(U^*) \geq \langle\Lambda^*, X_t^{k+1} + Y_t^{k+1} + Z_t^{k+1} - \bar{M}\rangle$$
$$\geq -\frac{1}{2}\left(\frac{1}{t}\|\Lambda^*\|^2 + t\|X_t^{k+1} + Y_t^{k+1} + Z_t^{k+1} - \bar{M}\|^2\right) \geq -\frac{1}{2t}(\|\Lambda^*\|^2 + \bar{c}_1), \tag{4.43}$$

where the second inequality is due to Cauchy-Schwarz inequality, and the last is due to (4.37). On the other hand, setting $W := W^*$ in (4.42), we obtain

$$\theta(U_t^{k+1}) - \theta(U^*) + \langle W_t^{k+1} - W^*, \Psi(W^*)\rangle \leq \frac{1}{t}\Theta(V^1, V^0, V^*).$$

Invoking the definition of $\Psi$ in (3.16c), we have

$$(W_t^{k+1} - W^*)^\top \Psi(W^*) = -\langle\Lambda^*, X_t^{k+1} + Y_t^{k+1} + Z_t^{k+1} - \bar{M}\rangle \geq -\frac{1}{2t}(\|\Lambda^*\|^2 + \bar{c}_1),$$

where the proof of the last inequality is similar to (4.43). Combining these two inequalities above, we get

$$\theta(U_t^{k+1}) - \theta(U^*) \leq \frac{1}{t}\Theta(V^1, V^0, V^*) + \frac{1}{2t}(\|\Lambda^*\|^2 + \bar{c}_1). \tag{4.44}$$

The inequalities (4.43) and (4.44) indicate that the assertion (4.38) holds by setting $\bar{c}_2 := \Theta(V^1, V^0, V^*) + \frac{1}{2}(\|\Lambda^*\|^2 + \bar{c}_1)$.