[Reviews · NeurIPS 2018]

Reviewer 1



This paper proposes a new kind of attack on recommender systems that the authors call unorganized malicious attacks. In these attacks, each item can only attacked by a limited number of users. The authors propose an algorithm for solving it and show global convergence of it. Finally, the authors evaluate their algorithm on two (semi-)synthetic tasks and one real-world one. My main issues with this paper are that (i) assumptions are not well motivated and justified, (ii) the theoretical results are not well aligned with the task of finding malicious users and (iii) the empirical evaluation does not have any recent baselines. With all the major revisions needed, I vote to reject this submission. Below are my detailed comments. Significance. The paper lacks proper motivation and justification for the scenario of malicious unorganized attacks that is studied in it. Why is this scenario more important or realistic than shilling attacks? To what degree does it generalize those? I think answering those questions right away in the introduction would strengthen the paper tremendously. Moreover, especially the formulation of the optimization problem (lines 148-159) needs proper justification. For example, why do we care about the rank function? Moving on to the theoretical results, I don't really feel there are connected well to the problem of finding malicious users, e.g., users with |Y_ij| != 0. The results give us convergence in terms of the Frobenius norm on Y, but I would like to see convergence in terms of the L_1 norm here. Quality. Regarding the empirical evaluation, I am not convinced that the baselines offer a fair comparison. Especially the N-P and k-means based methods are very simple. The most recent baseline is 5 years old which would need some justification. I would also expect to see Maximum-margin matrix factorization as a baseline here. Originality. This work is based on standard approaches for matrix completion and convex analysis. The main novelty is the introduction of the attack matrix Y. Hence, I would consider the work in this paper to be incremental overall. Clarity. I had a hard time reading through the paper because relevant information was scattered throughout and it also lacked a clear focus. I would rewrite the paper so that it is clear from the very begining that the problem you are trying to solve is to find users that were part of an unorganized malicious attack. This will make the theory and introduction easier to write. Right now, it's not clear what exactly the problem is that the authors seek out to solve. For example, in the introduction the paper says "in this paper, we investigate a new attack model" -- yes, but what is the goal? There are also various typos ("recommender system has" -> "recommender systems have", "there are many works" -> "there is much work", "is a (or partial) matrix" -> "is a partial matrix" etc.) that should be fixed. Moreover, the first paragraph in the introduction is unnecessarily broad and should be more focused, then the last paragraph can be omitted since it is already clear from the section headings what follows. The problem formulation should follow right after the introduction, I would move related work to the end. I would also start with the new unorganized attack scenario and then later talk about the other attack strategies. Section 3 should also be written to make the assumptions and model more prominent. Author rebuttal: Thank you for clarifying my concerns. However, the revisions needed / proposed would be substantial, and the paper would need another round of reviewing. Hence, I still vote to reject this paper.

Reviewer 2



Summary: This paper considers a new attack style named unorganized malicious attacks in which attackers individually utilize a small number of user profiles to attack different items without any organizer. The authors formulate this kind of attacks and give a clear comparison between unorganized malicious attacks and shilling attacks. Detecting this kind of attacks is an interesting and important problem. A framework is established where the observed rating matrix is the sum of the ground-truth rating matrix X, a sparse attack matrix Y and a Gaussian noise matrix Z. The goal is to accurately recover X and Y to determine who are malicious users, which is achieved by solving a variant of matrix completion problem. Theorems 1 and 2 give the theoretical guarantee about the recovery of X and Y under some mild assumptions. The optimization problem is solved by alternating splitting augmented Lagrangian method. Theorem 3 guarantees the global convergence of this method and this theoretical result for three-block convex problems is attractive. The experiments are conducted on the common-used datasets MovieLens where attack profiles are inserted by the authors and a real dataset Douban10K. Empirical studies show that the proposed method UMA outperforms other state-of-the-art methods for attack detection and robust PCA. Detailed comments: I do agree that unorganized attacks detection is a well-motivated and important problem, while most attack detection methods focus on shilling attacks. Considering to address this problem is itself a novel contribution. The paper is clearly written. It is natural and reasonable to formulate the unorganized malicious attacks detection as a variant of matrix completion problem and each term of the optimization problem is suitable. Different from robust PCA, it considers the specific characteristic of unorganized attacks to better distinguish Y and Z. The global convergence of the three-block convex problem is also attractive. Various state-of-the-art attack detection methods are compared on the common data set MovieLens. The real data set Douban further validates the effectiveness of their method. Results are well analyzed and I am most impressed by the analysis of the failure of the traditional attack detection approaches and the advantages of their method against robust PCA and MF-based method. Minor suggestions: What is the intuitive explanations for the assumption about X_0 (Eq. 7)? Besides, I think it will be a good extension of this work to further consider side information.

Reviewer 3



This work focuses on a new type of attack in collaborative filtering named “unorganized malicious attacks”, and the uncoordinated attacks are different from shilling attacks where attack profiles are produced in an organized way. Attack detection based on the common characteristics of the attack profiles may not work in the new attack style. The authors propose a new optimization function and algorithm UMA which works to distinguish attacks from noise under the low-rank assumption. They also give the theoretical guarantees of recovery and global convergence. Empirical results show the effectiveness of the proposed method and some insightful discussions are given. This paper is well structured and clearly written. Unorganized malicious attacks detection is a well-motivated problem and I personally find it interesting. Such uncoordinated attacks do exist in the common recommendation systems. For example, on Amazon, online sellers may produce several fake profiles to give negative ratings to their competitors’ good products and the negative ratings do affect customers’ choices. Various works has been conducted to study attack detection in collaborative filtering, while most works focus on coordinated shilling attacks and detect attacks by finding the group with common characteristics which are different from normal profiles. Detecting unorganized attacks is a hard problem. The authors propose a new model to distinguish sparse attacks from noises under the low-rank assumption. Instead of finding groups of profiles acting together, they directly find the malicious ratings that deviate from the real ratings a lot. Each term in the formulation is reasonable and the whole formulation is precise. Is that possible to use L2,1-norm to regularize Y instead L1-norm? What is the reason that this problem prefers the L1-norm? Experiments show the effectiveness of the proposed method, and I am impressed by the hard case under general unorganized malicious attacks. This case considers that attack profiles can come from normal profiles with a malicious rating. UMA performs consistently well to detect these attacks, while the comparison methods suffer performance degradation. Detailed explanations about these phenomena should be added. ==========Update after authors' response========== I have read authors' response. The response has addressed my concerns.